# Cognate Transformer for Automated Phonological Reconstruction and Cognate Reflex Prediction

**V.S.D.S.Mahesh Akavarapu** and **Arnab Bhattacharya**
Dept. of Computer Science and Engineering
Indian Institute of Technology Kanpur
maheshak@cse.iitk.ac.in, arnabb@cse.iitk.ac.in

## Abstract

*Phonological reconstruction* is one of the central problems in historical linguistics where a proto-word of an ancestral language is determined from the observed cognate words of daughter languages. Computational approaches to historical linguistics attempt to automate the task by learning models on available linguistic data. Several ideas and techniques drawn from computational biology have been successfully applied in the area of *computational historical linguistics*. Following these lines, we adapt MSA Transformer, a protein language model, to the problem of *automated phonological reconstruction*. MSA Transformer trains on multiple sequence alignments as input and is, thus, apt for application on aligned cognate words. We, hence, name our model as *Cognate Transformer*. We also apply the model on another associated task, namely, *cognate reflex prediction*, where a reflex word in a daughter language is predicted based on cognate words from other daughter languages. We show that our model outperforms the existing models on both tasks, especially when it is pre-trained on masked word prediction task.

## 1 Introduction

*Phonological reconstruction* of a word in an ancestral proto-language from the observed cognate words, i.e., words of supposed common origin, in the descendant languages is one of the central problems in *historical linguistics*, a discipline that studies diachronic evolution of languages (Campbell, 2013). For example, the cognate words French *enfant*, Spanish *infantes* and Italian *infanti* all trace to the proto-form *infantes* in Latin meaning 'children', which is an attested language in this case. In most cases, the proto-language is not attested and has to be rather reconstructed. The process of arriving at such phonological reconstruction usually involves multiple steps including gathering potential cognate words, identifying systematic sound cor-

respondences, and finally reconstructing the proto-phonemes. This procedure is known as the 'comparative method' (Ringe and Eska, 2013), which is traditionally carried out manually.

Several *automated phonological reconstruction* algorithms emerged in the last decade. Some of these are drawn or inspired from computational biology, for example, Bouchard-Côté et al. (2013). In general, computational historical linguistics draws techniques such as sequence alignment and phylogenetic inference from computational biology, in addition to the techniques known from historical linguistics and computational linguistics or natural language processing (Jäger, 2019). On similar lines, we adapt the MSA transformer, introduced in Rao et al. (2021) for modeling multiple sequence alignment (MSA) protein sequences, for the problem of phonological reconstruction which takes as input a cognate word set in the form of MSAs. Henceforth, we name the model introduced here as *Cognate Transformer*.

We also apply our model on *cognate reflex predicion* task, where an unknown, i.e., a masked reflex in a daughter language is to be predicted based on the attested reflexes from other daughter languages (List et al., 2022b). For instance, in the previous example, if we mask French *enfant*, the task would involve arriving at the word form correctly based on Spanish *infantes* and Italian *infanti*. One can notice that this task can serve as a pre-training objective for the proto-language reconstruction task described previously. Hence, we also pre-train the Cognate Transformer on the cognate reflex prediction task.

Further, most of the existing models are fitted on a per language family basis, i.e., on one dataset at a time consisting of a single language family. Thus, the utility of either transfer learning or simultaneous fitting across several language families has not yet been demonstrated. This is desirable even from the linguistic perspective since it is well known

that the sound changes are phonologically systematic and, thus, often similar sound changes operate across different language families (Campbell, 2013). For instance, the sound change involving palatalization of a velar consonant say /k/ > /tʃ/ can be observed in the case of Latin *caelum* /kaɪlʊm/ to Italian *cielo* /tʃɛːlo/ as well as in the supposed cognate pairs *cold* versus *chill*, which is a reminiscence of historical palatalization in Old English.[1] Hence, owing to the presence of commonalities across language families in terms of sound change phenomena, training models simultaneously across multiple language families should be expected to yield better results than when training on a single language family data at a time. This is well reflected in our present work.

## 1.1 Problem Statements

There are two tasks at hand as mentioned before, namely, *cognate reflex prediction* and *proto-language reconstruction*.

An input instance of the cognate reflex prediction task consists of a bunch of cognate words from one or more related languages with one language marked as unknown; the expected output would be the cognate reflex in that particular language which is marked unknown. An example from the Romance languages is:

*Input:* [French] ʒ ə n j ɛ v ʁ, [Portuguese] ?, [Italian] dʒ i n e p r o

*Output:* [Portuguese] ʒ u n i p i r ʊ

The input for the proto-language reconstruction task consists of cognate words in the daughter languages and the expected output is the corresponding word in the ancestral (proto-) language. We model this as a special case of cognate reflex prediction problem where the proto-language is always marked as unknown. For instance, in the above example, Latin would be marked as unknown:

*Input:* [Latin] ?, [French] ʒ ə n j ɛ v ʁ, [Portuguese] ʒ u n i p i r ʊ

*Output:* [Latin] j uː n ɪ p ɛ r ʊ m

## 1.2 Contributions

Our contributions are summarized as follows. We have designed a new architecture, Cognate Transformer, and have demonstrated its efficiency when applied to two problems, namely, proto-language reconstruction and cognate reflex prediction, where it performs comparable to the existing methods. We have further demonstrated the use of pretraining in proto-language reconstruction, where the pre-trained Cognate Transformer outperforms all the existing methods.

The rest of the paper is organized as follows. Existing methodologies are outlined in §2. The workflow of Cognate Transformer is elaborated in §3. Details of experimentation including dataset information, model hyperparameters, evaluation metrics, etc. are mentioned in §4. Results along with discussions and error analysis are stated in §5.

## 2 Related Work

Several methods to date exist for proto-language reconstruction, as mentioned previously. We mention a notable few. Bouchard-Côté et al. (2013) employs a probabilistic model of sound change given the language family's phylogeny, which is even able to perform unsupervised reconstruction on Austronesian dataset. Ciobanu and Dinu (2018) performed proto-word reconstruction on Romance dataset using conditional random fields (CRF) followed by an ensemble of classifiers. Meloni et al. (2021) employ GRU-attention based neural machine translation model (NMT) on Romance dataset. List et al. (2022a) presents datasets of several families and employs SVM on trimmed alignments.

The problem of cognate reflex prediction was part of SIGTYP 2022 shared task (List et al., 2022b), where the winning team (Kirov et al., 2022) models it as an image inpainting problem and employs a convolutional neural network (CNN). Other high performing models include a transformer model by the same team, a support vector machine (SVM) based baseline, and a Bayesian phylogenetic inference based model by Jäger (2022). Other previous approaches include sequence-to-sequence LSTM with attention, i.e., standard NMT based (Lewis et al., 2020) and a mixture of NMT experts based approach (Nishimura et al., 2020).

The architecture of MSA transformer is part of Evoformer used in AlphaFold2 (Jumper et al., 2021), a protein structure predictor. Pre-training of MSA transformer was demonstrated by Rao et al. (2021). Handling MSAs as input by using 2D convolutions or GRUs was demonstrated by Mirabello and Wallner (2019) and Kandathil et al. (2022).

---

[1] For International Phonetic Alphabet (IPA) notation, see https://en.wikipedia.org/wiki/International_Phonetic_Alphabet.

| [French] | ʒ | ə | n | j | ɛ | v | - | ʁ | - | - |
|---|---|---|---|---|---|---|---|---|---|---|
| [Italian] | dʒ | i | n | - | e | p | - | r | o | - |
| [Spanish] | x | u | n | - | i | p | ɾ | o | - |
| [Latin] | j | uː | n | - | ɪ | p | ɛ | r | ʊ | m |

Table 1: Aligned phoneme sequences

| [French] | ʒ | ə | n | j.ɛ | v | - | ʁ | - |
|---|---|---|---|---|---|---|---|---|
| [Italian] | dʒ | i | n | e | p | - | r | o |
| [Spanish] | x | u | n | i | p | - | ɾ | o |
| [Latin] | ? | ? | ? | ? | ? | ? | ? | ? |
| [Latin] | j | uː | n | ɪ | p | ɛ | r | ʊ.m |

Table 2: Trimmed input and output alignments

## 3 Methodology

In this section, the overall workflow is described. The input phoneme sequences are first aligned (§3.1), the resulting alignments are trimmed (§3.2), and then finally passed into the MSA transformer with token classification head (§3.3). In the training phase, the output sequence is also aligned while in the testing phase, trimming is not performed. The first two steps are the same as described in List et al. (2022a) and are briefly described next.

### 3.1 Multiple Sequence Alignment

The phenomenon of sound change in spoken languages and genetic mutations are similar. As a result, multiple sequence alignment and the methods surrounding it are naturally relevant here as much as they are in biology. The phonemes of each language in a single cognate set are aligned based on the sound classes to which they belong. An example of an alignment is given in Table 1.

We use the implementation imported from the library lingpy (List and Forkel, 2021) which uses the sound-class-based phonetic alignment described in List (2012). In this algorithm, the weights in pairwise alignments following Needleman and Wunsch (1970) are defined based on the sound classes into which the phonemes fall. Multiple sequences are aligned progressively following the tree determined by UPGMA (Sokal and Michener, 1975).

### 3.2 Trimming Alignments

In the example given in Table 1, one can observe that during testing, the final gap (hyphen) in the input languages (i.e., excluding Latin) will not be present. Since the task is essentially a token classification, the model will not predict the final token 'm' of Latin. To avoid this, alignments are trimmed as illustrated in Table 2 for the same example.

This problem is discussed in detail in (List et al., 2022a) and the solution presented there has been adopted here. In particular, given the sequences to be trimmed, if in a site all tokens are gaps except in one language, then that phoneme is prepended

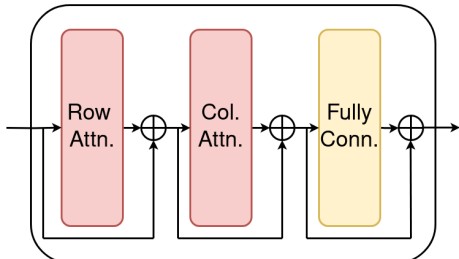

Figure 1: A single layer of MSA Transformer.

to the following phoneme with a separator and that specific site is removed. For the last site, the lone phoneme is appended to the penultimate site. Following (List et al., 2022a), trimming is skipped for testing as it has been observed to cause a decrease in performance. The reason for this is mentioned in (Blum and List, 2023). Briefly stating it, gaps in daughter languages can point to a potential phoneme in the proto-language. While training however, they are redundant and can be trimmed since proto-language is part of alignment.

### 3.3 MSA Transformer

The MSA Transformer, proposed in (Rao et al., 2021), handles two-dimensional inputs with separate row and column attentions (each with multiple heads) in contrast with the usual attention heads found in standard transformer architectures (Vaswani et al., 2017). It uses learned positional embeddings only across rows since a group of rows does not make up any sequential data. The outputs of row attentions and column attentions are summed up before passing into a fully connected linear layer (see Figure 1). MSA Transformer, despite its name, is not an encoder-decoder transformer but rather only an encoder like BERT (Devlin et al., 2018), except with the ability to handle 2D input (see Figure 2).

### 3.4 Workflow

The aligned input sequences thus trimmed are passed into MSA Transformer as tokens. A single input instance to an MSA Transformer is a 2D array of tokens. The overall architecture of the Cognate Transformer is illustrated in Figure 2. Due to trim-

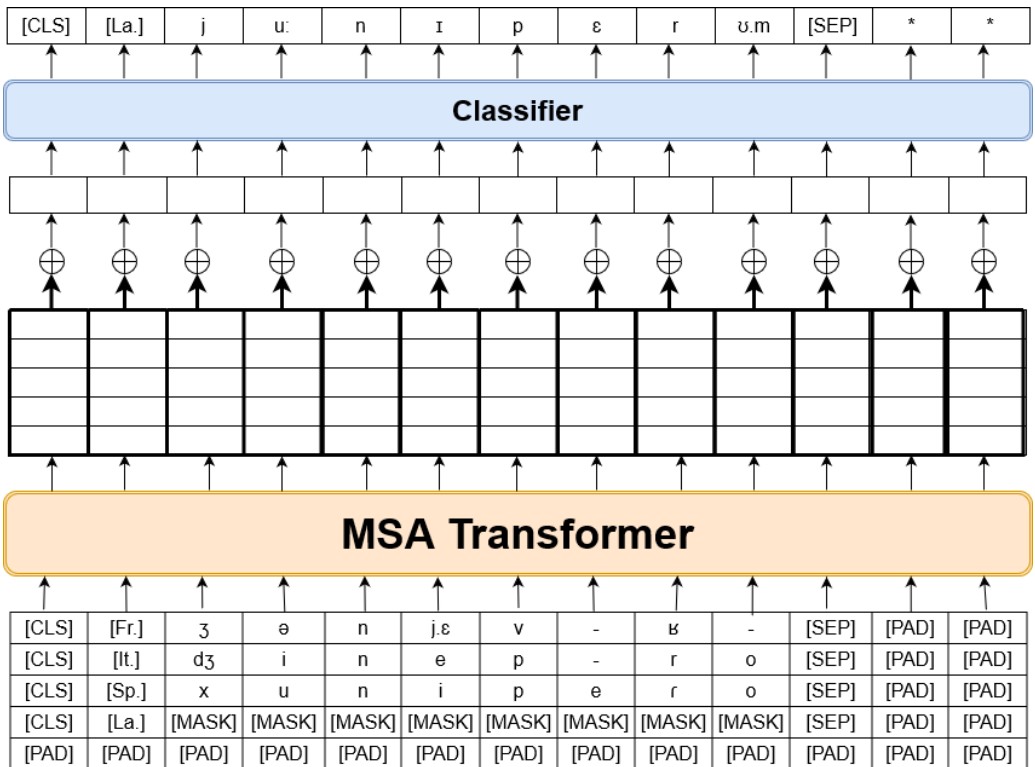

Figure 2: Cognate Transformer architecture: an input instance is passed into an MSA transformer, where the resultant embeddings are summed and normalized along columns, which are then finally passed into a classifier.

ming, several phonemes can be joined together as one token. Hence, with trimming the total number of tokens or the vocabulary size can be above 1000 or even 2000 based on the training dataset, while without trimming the vocabulary size would essentially be close to the total number of phonemes possible which would be only a few hundreds.

Meloni et al. (2021) incorporate the information regarding the language of a word through a language embedding concatenated to the character/token embedding. We instead treat *language information* as a separate token attached to the beginning of the phoneme sequence. Use of language embeddings with transformer based models was initially present in the multi-language model XLM (Conneau and Lample, 2019). It was however discontinued in the later versions(Conneau et al., 2020). We similarly have decided to remove the language embedding and instead use a special token denoting language as it is less complex in implementation. Other special tokens used include the usual [CLS] to mark the beginning, [SEP] to mark the ending of a word, [PAD] for padding, and [MASK] to replace '?' in the unknown word (see Table 2) or the word to be predicted. Thus, the input batch padded appropriately is passed on to the MSA Transformer.

The normal output of an MSA Transformer is a 2D array of embeddings per instance. To this, we add an addition layer that sums over columns to give a 1D array of embeddings per instance as output. In other words, if the overall dimensions of the MSA transformer output were (`batch_size` × `num_languages` × `msa_length` × `hidden_size`) then, for our case, the final dimensions after summing up along columns are (`batch_size` × `msa_length` × `hidden_size`). To this, we add a normalizer layer followed by a classifier, i.e., a linear layer followed by cross-entropy loss. This is illustrated in Figure 2.

### 3.5 Pre-training

The described model can support pre-training in a form similar to masked language modeling where a word from a cognate set is entirely masked but the language token remains unmasked corresponding to the language that is to be predicted. In other words, *pre-training* is the same as training for cognate prediction task. For proto-language reconstruction, however, pre-training can be done. As a result, we pre-train Cognate Transformer on the data of the cognate reflex prediction task. It is further *fine-tuned* on the proto-language reconstruction task.

We have used the publicly available implementa-

| Family | Lngs. | Words | Cogs. |
|---|---|---|---|
| **Training data** | | | |
| Tshanglic | 8 | 2063 | 403 |
| Bai | 9 | 5773 | 969 |
| Sino-Tibetan | 7 | 1426 | 248 |
| Sui | 16 | 10139 | 1048 |
| Uto-Aztecan | 9 | 771 | 118 |
| Afro-Asiatic | 19 | 2583 | 340 |
| Dogon | 16 | 4405 | 971 |
| Japonic | 10 | 1802 | 278 |
| Indo-European | 4 | 1320 | 512 |
| Burmish | 7 | 2501 | 576 |
| | | **32783** | **5463** |
| **Surprise data** | | | |
| Atlantic-Congo | 10 | 1218 | 388 |
| Hui | 19 | 9750 | 518 |
| Chapacuran | 10 | 939 | 187 |
| Western Kho-Bwa | 8 | 5214 | 915 |
| Berta | 4 | 600 | 204 |
| Palaung | 16 | 1911 | 196 |
| Burmish | 9 | 2202 | 467 |
| Indo-European | 5 | 565 | 212 |
| Karen | 8 | 2363 | 379 |
| Bai | 10 | 4356 | 658 |
| | | **29118** | **4124** |

Table 3: Dataset for reflex prediction task

| Family | Lngs. | Words | Cogs. |
|---|---|---|---|
| Bai | 10 | 459 | 3866 |
| Burmish | 9 | 269 | 1711 |
| Karen | 11 | 365 | 3231 |
| Lalo (Yi) | 8 | 1251 | 7815 |
| Purus | 4 | 199 | 693 |
| Romance | 6 | 4147 | 18806 |
| | | **6690** | **36122** |

Table 4: Dataset for Proto-language reconstruction task

tion of MSA transformer by the authors[2], on top of which we added the layers required for the Cognate Transformer architecture. We have used tokenization, training, and related modules from HuggingFace library (Wolf et al., 2020). The entire code is made publicly available[3].

## 4 Experimental Setup

### 4.1 Datasets

We use the SIGTYP 2022 dataset (List et al., 2022b) for the cognate reflex prediction task. It consists of two different subsets, namely, training and surprise, i.e., evaluation data from several language families. The statistics for this dataset is provided in Table 3. Surprise data was divided into different test proportions of 0.1, 0.2, 0.3, 0.4, and 0.5 for evaluation. Among these, we only report for the test proportions 0.1, 0.3, and 0.5.

For the proto-language reconstruction task, the dataset provided by List et al. (2022a) is used. It consists of data from 6 language families, namely, Bai, Burmish, Karen, Lalo, Purus, and Romance whose statistics are listed in Table 4. This is divided

into test proportion 0.1 by List et al. (2022a). We further test for proportions 0.5 and 0.8. For pre-training the Cognate Transformer for this task, we use the entire training data of both the tasks with words from proto-languages removed.

### 4.2 Model Hyperparameters

We have tested two variations of the proposed Cognate Transformer architecture, namely *CogTran-tiny* and *CogTran-small*. CogTran-tiny has hidden size 128, intermediate size 256, 2 attention heads, and 2 layers with overall 1 million parameters. CogTran-small has hidden size 256, intermediate size 512, 4 attention heads, and 4 layers with overall 4.4 million parameters. Both models have a vocabulary size of about 2,300.

For pre-training, only CogTran-small is used, since it consistently outperforms CogTran-tiny. The training is carried out with 48 epochs for pre-training, with 9 epochs for finetuning in the proto-language reconstruction task, 24 epochs for non-pre-trained in the same task, and 32 epochs for cognate-reflex prediction task, using Adam optimizer with weight decay (Loshchilov and Hutter, 2017) as implemented in HuggingFace transformers library (Wolf et al., 2020) with learning rate 1e-3 and batch size of 64. For finetuning the pre-trained model, the batch size is 48.

### 4.3 Evaluation

We use the metrics *average edit distance (ED)*, *average normalized edit distance (NED)*, and *B-Cubed F1 score (BC)* following List et al. (2022a) for evaluating the models. Edit distance is the well-known Levenshtein distance (Levenshtein, 1965), both with or without normalization by the lengths of the source and target strings being compared. B-Cubed F1 score (Amigó et al., 2009) was applied to phoneme sequences by List (2019), where similarity is measured between aligned predicted and gold sequences. B-Cubed F1 score measures

[2]https://github.com/facebookresearch/esm
[3]https://github.com/mahesh-ak/CognateTransformer

the similarity in the structures and, hence, in the presence of systematic errors, carries less penalty than edit distance. As (normalized) edit distance is a distance measure, the lower the distance, the better the model. On the other hand, for B-Cubed F1 it is opposite, i.e., the higher the score, the better the model. We import the metric functions from the `LingRex` package (List and Forkel, 2022).

## 4.4 Methods for Comparison

The results of the cognate reflex prediction task are compared directly against those of the top performing model in the SIGTYP 2022 task – Kirov et al. (2022). Here, direct comparison between the models is possible since the datasets including the test divisions are the same.

However, for the proto-language reconstruction task, the previous state-of-the-art model (Meloni et al., 2021) reports only on the Romance dataset with test proportion 0.12 and the baseline SVM model (List et al., 2022a) with additional features such as position, prosodic structure, etc., marked as SVM+PosStr is tested only with test proportion 0.1. However, the code is openly provided for the SVM-based model and, hence, results were generated for other test proportions 0.5 and 0.8 as well.

To compare the results of proto-language reconstruction with the NMT model given by Meloni et al. (2021) for which the code is not publicly available, we build a best-effort appropriate model identical to the one described there with 128 units Bidirectional GRU encoder followed by same sized GRU decoder followed by attention and linear layer with dimension 256 followed by a classifier. The input is encoded as a 96-dimensional embedding for each token concatenated with 32-dimensional language embedding. The training parameters are the same as previously stated in §4.2 except that the number of epochs trained is 32 and the batch size is 16. For the Romance data part, the results obtained are ED 1.287 and NED 0.157 whereas those reported by Meloni et al. (2021) for Romance data (IPA) with almost similar test proportion (0.12) are ED 1.331 and NED 0.119. Thus, the edit distances match whereas normalized ones do not. We speculate that the NED reported by Meloni et al. (2021) could be erroneous due to possible inclusion of delimiter while calculating the length of the strings, since by (mis)considering delimiters, we obtain a similar NED 0.121 for the model we train. This can be confirmed by observing the ED-to-NED propor-

tions of the corresponding scores obtained by the SVM-based model for the Romance dataset: ED 1.579 and NED 0.190, which we generate using the code made available by List et al. (2022a). Alternatively, the disparity in NED could also be attributed to differences in the sizes of the dataset used for training. However, it is unclear how agreement in ED score could have been then possible. Due to absence of both appropriate model and data, we assume that the NMT model we have built is a good reproduction of that built by Meloni et al. (2021).

All models compared in the proto-language reconstruction task are 10-fold cross-validated.

# 5 Results

In this section, we present and discuss in detail the results of our Cognate Transformer and other state-of-the-art models on the two tasks.

## 5.1 Cognate Reflex Prediction

The results of the cognate reflex prediction task are summarized in Table 5. The edit distance (ED), normalized edit distance (NED), and B-Cubed F1 (BC) scores are provided for Cognate Transformer across the test proportions 0.1, 0.3, and 0.5 along with the best performing model of the SIGTYP 2022 (List et al., 2022b) task, namely, the CNN inpainting (Kirov et al., 2022). CogTran-small consistently outperforms the previous best models across all test proportions. In particular, the difference in scores between Cognate transformer and the CNN inpainting model becomes prominent with increasing test proportion. Hence, it can be concluded here that Cognate Transformer is more robust than other models. The language family wise results for the best performing model, CogTran-small, are provided in Appendix A.

## 5.2 Proto-Language Reconstruction

The results of the proto-language reconstruction task are summarized in Table 6 with the same evaluation metrics along with comparisons with other previously high performing models, namely, SVM with extra features by List et al. (2022a) and NMT (GRU-attention) based by Meloni et al. (2021) for the test proportions 0.1, 0.5, and 0.8. Previously, there were no comparisons between SVM-based and NMT-based models. Here, we find that the SVM-based model performs consistently better than the NMT-based model. In other words, the GRU-Attention-based NMT model does not appear

| Test proportion | Method | ED | NED | BC |
|---|---|---|---|---|
| | CogTran-tiny | 1.0901 | 0.2997 | 0.7521 |
| 0.1 | CogTran-small | **0.8966** | **0.2421** | **0.7823** |
| | Mockingbird - Inpaint (Kirov et al., 2022) | 0.9201 | 0.2431 | 0.7673 |
| | CogTran-tiny | 1.3223 | 0.3497 | 0.6612 |
| 0.3 | CogTran-small | **1.1235** | **0.2919** | **0.6954** |
| | Mockingbird - Inpaint (Kirov et al., 2022) | 1.1762 | 0.2899 | 0.6717 |
| | CogTran-tiny | 1.4521 | 0.3873 | 0.6257 |
| 0.5 | CogTran-small | **1.2786** | **0.3332** | **0.6477** |
| | Mockingbird - Inpaint (Kirov et al., 2022) | 1.4170 | 0.3518 | 0.6050 |

Table 5: Cognate reflex prediction results.

| Test proportion | Method | ED | NED | BC |
|---|---|---|---|---|
| | CogTran-tiny | 0.8081 | 0.1760 | 0.7946 |
| | CogTran-small | 0.7772 | 0.1683 | 0.7968 |
| 0.1 | CogTran-small Pretrained | **0.7459** | **0.1595** | **0.8081** |
| | SVM + PosStr (List et al., 2022a) | 0.7612 | 0.1633 | 0.8080 |
| | NMT GRU + Attn. (Meloni et al., 2021) | 1.0296 | 0.1909 | 0.7560 |
| | CogTran-tiny | 0.9013 | 0.1966 | 0.7279 |
| | CogTran-small | 0.8750 | 0.1899 | 0.7330 |
| 0.5 | CogTran-small Pretrained | **0.8177** | **0.1760** | **0.7534** |
| | SVM + PosStr (List et al., 2022a) | 0.8455 | 0.1839 | 0.7425 |
| | NMT GRU + Attn. (Meloni et al., 2021) | 1.2585 | 0.2362 | 0.6733 |
| | CogTran-tiny | 1.1043 | 0.2455 | 0.6781 |
| | CogTran-small | 1.0697 | 0.2359 | 0.6817 |
| 0.8 | CogTran-small Pretrained | **0.9754** | **0.2142** | **0.7132** |
| | SVM + PosStr (List et al., 2022a) | 1.0630 | 0.2391 | 0.6800 |
| | NMT GRU + Attn. (Meloni et al., 2021) | 1.8640 | 0.3546 | 0.5538 |

Table 6: Proto-language reconstruction results.

to scale well in harder situations, i.e., for higher test proportions when compared with the other models. While CogTran-small achieves results similar to the SVM-based models, pre-training makes a difference. The pre-trained Cognate transformer outperforms all the other models in all test proportions. Although the increase in the proportion 0.1 is not much significant, paired t-test between best performing model and the next best model i.e. CogTran-small Pretrained and SVM-based yield significance of $p < 0.01$ in low-resource proportions i.e. 0.5 and 0.8 . The language family wise results and standard deviations for the best performing model, CogTran-small Pretrained are provided in Appendix B. Note that SVM-based model was also part of SIGTYP 2022 (List et al., 2022b) where it lags well behind CNN inpainting model. Hence, cognate transformer generalizes well across tasks hence gains from architecture are obvious.

### 5.3 Error Analysis

To analyze errors, we consider the pre-trained and finetuned CogTran-small on the proto-language reconstruction task for the easiest and hardest test proportions 0.1 and 0.8 over fixed data (without cross-validation). Figure 3 shows the 30 most common sound exchange errors by the models. An example of sound exchange error, u/a means either 'a' is predicted in place of 'u' or vice versa. To make this plot, we first gather the frequencies of sound exchanges for the various language families in data by comparing the aligned predicted and gold reconstructions. These frequencies are normalized for each proto-language or language family and finally combined and normalized again. Normalization at the language family level is important since few language families show more tendencies for certain types of errors than others. Since data is not equally available from all families, a language family with

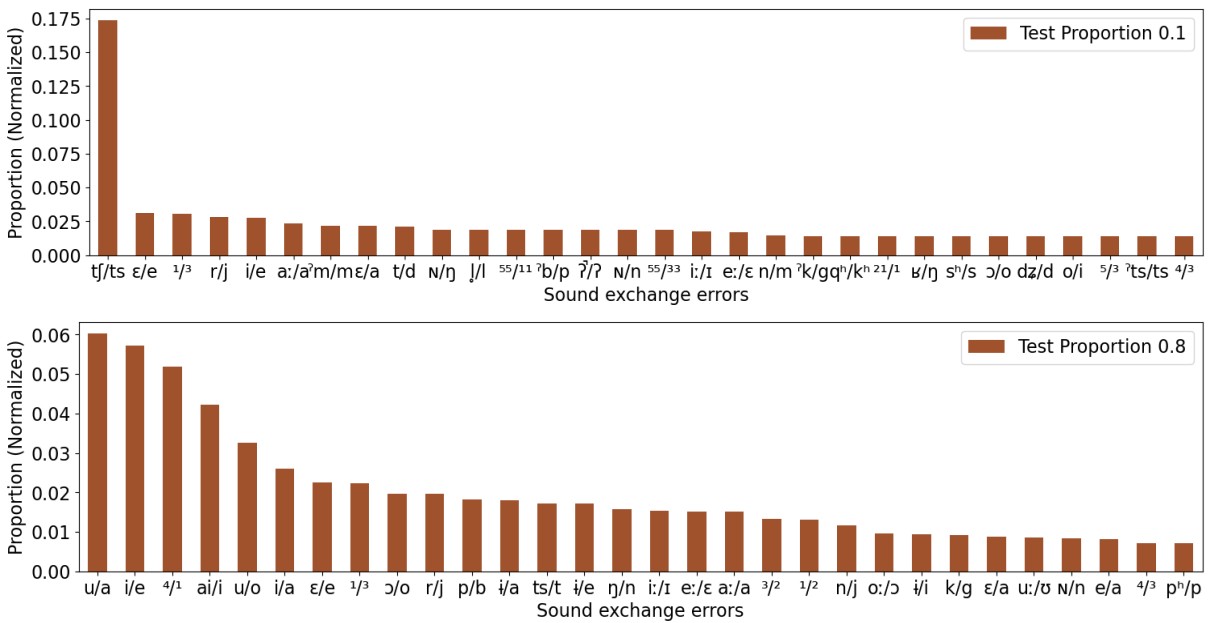

Figure 3: Top-30 most common sound exchange errors out of over 400 errors for pre-trained CogTran-small on proto-language reconstruction task with test proportions 0.1 (top) and 0.8 (bottom).

more data influences the outcome. For example, among the datasets used for the task, the Romance dataset comprises half of them. We observe that Romance data shows more vowel-length-related errors as also observed by Meloni et al. (2021) and, thus, proportion of such errors is inflated. Hence, normalization is carried out at the language family level to prevent such biases. We normalize per family by dividing the frequency of a particular error type in a family by the total number of errors in that family. Normalized frequencies thus obtained per error type per family are combined by adding up across families and then normalized again.

The most frequent sound exchange errors are plotted in Figure 3 which make up respectively, for test proportions 0.1 and 0.8, about 71% and 60% of total such errors. One can observe from the plot that the most common vowel errors are the exchange of short vowels /u/ and /i/ with a neutral vowel /a/, vowel raising-lowering, i.e., exchange of /i/ ∼ /e/, /u/ ∼ /o/, diphthong-monophthong exchanges /ai/ ∼ /i/, tense-laxed exchanges, i.e., /ɛ/ ∼ /e/ and /ɔ/ ∼ /o/. Vowel length confusions, i.e., /iː/ ∼ /ɪ/, /eː/ ∼ /e/, /aː/ ∼ /a/, /oː/ ∼ /ɔ/, /uː/ ∼ /ʊ/ also make up a significant portion. Overall, vowel/consonant length errors make up to about 10% sound exchange errors each in both cases. Among consonant errors, one can observe voiced-unvoiced or glottalized-unglottalized consonant exchanges like /p/ ∼ /b/, /ˀk/ ∼ /g/, aspiration errors, i.e., /pʰ/ ∼ /p/, /tʰ/ ∼ /t/, change of place of articu-

lation like /ŋ/ ∼ /n/, /s/ ∼ /h/, etc. Tone exchange errors like /¹/ ∼ /³/ also exist which contribute to about 10% in each of the cases. Affricatives exchange error /tʃ/ ∼ /ts/ appears prominently in the case of test proportion 0.1. Overall, these are the most general kinds of errors; however, exact types of errors are observed to be dependent on the language family. Hence, although most general ones are universally observed, significant differences can be expected based on the particular datasets.

## 5.4 Zero-shot Attempt

Previously, we discussed the results of proto-language reconstruction for various test proportions. Among these, the highest proportion considered, i.e., 0.8, can be thought of as a *few-shot* learning case, since for some of the language families like Purus and Burmish, the number of training instances, i.e., cognate sets is less than 50. We next consider the pre-trained model for the same task without any finetuning; in other words, we consider the *zero-shot* case. The scores achieved by such a model are 2.6477 ED, 0.5758 NED, and 0.5499 BC, which means that more than 40% of a word on average in generated reconstructions are correct. An example input instance and its corresponding output and gold data from the Romance dataset:

*Input:* [Latin] ?, [French] p ɛ ʁ s p i k y i t e, [Italian] p e r s p i k u i t a

*Output:* [French] p e r s p i k y i t a

*Gold:* [Latin] p ɛ r s p ɪ k ʊ ɪ t aː.t.ɛ.m

In the above example, the output language token is incorrect. Since the proto-languages (in this case, Latin) have been excluded entirely in pre-training, this can be expected. One can also observe that the output word completely agrees with neither Italian nor French, although the inclination is more toward the former (with a ED of 1). A similar observation was made by Meloni et al. (2021) where the network attended most to Italian since it is conservative than most other Romance languages.

### 5.5 Learned Sound Changes

Here, we consider the finetuned pre-trained model on the proto-language reconstruction task to observe the learned sound changes by the network in the hardest scenario, i.e., with test proportion 0.8. The following example reveals an instance where palatalization appearing in Romance languages is correctly reconstructed to a non-palatal consonant:

*Input:* [Latin] ?, [French] s j ɛ, [Spanish] θ j e
*Output:* [Latin] k ɛ

We now consider *metathesis*, a non-trivial complex sound change where positions of phonemes are interchanged. The following example is from the training set which the network learns correctly and demonstrates the metathesis *-bil- > -ble-*.

*Input:* [Latin] ?, [French] ɛ̃ p ɛ ʁ s ɛ p t i b l, [Spanish] i m p e ɾ θ e pː t i β l e
*Output:* [Latin] ɪ m p ɛ r k ɛ p t ɪ b ɪ l ɛ m

Following is an example from the test set where the model confuses a complex metathesis pattern occurring in Hispano-Romance, *-bil- > -lb-*.

*Input:* [Latin] ?, [Spanish] s i l β a r, [French] s y b l e, [Portuguese] s i l v a ɹ
*Output:* [Latin] s y b l w aː r ɛ
*Gold:* [Latin] s iː b ɪ l aː r ɛ

Even the model finetuned on test proportion 0.1 does not get this example correct. Its output is

*Output:* [Latin] s y b l ɔ aː r ɛ

Hence, metathesis can be seen as a hard sound change to be learned by this model. This is not surprising since metathesis or site exchange does not naturally fit into the sequence alignment approach which fundamentally only models insertions and deletions at any site. Thus, it is worthwhile to investigate more on this aspect by training the network on language families that exhibit systematic metathesis to understand its behavior.

### 6 Conclusions

In this paper, we adapted MSA transformer for two phonological reconstruction tasks, namely, cognate reflex prediction and proto-language reconstruction. Our novel architecture, called Cognate Transformer, performs either comparable to or better than the previous methods across various test-train proportions consistently. Specifically, the pre-trained model outperforms the previous methods by a significant margin even at high test-train proportions, i.e., with very less trainable data reflecting a more realistic scenario.

To the best of our knowledge, this work demonstrates the utility of transfer learning when applied to historical linguistics for the first time. In this paper, the data is in IPA representation, but this is not necessary as long as words can be aligned with properly defined sound classes in the respective orthographic representations. Thus, relaxing the IPA input constraint can increase the amount of trainable data and pre-training with more data would most likely improve the performance of not only the problem of automated phonological reconstruction but can be demonstrated in the future for an important related task, namely automated cognate word detection. Further, more standard ways of pre-training such as masking only a couple of tokens across all languages instead of a complete word of a single language can be adapted in future.

### Limitations

In the task of proto-language reconstruction, it can be seen from the results (Table 6) that CogTrans-small i.e. the plain Cognate Transformer model without pre-training slightly underperforms the SVM-based model at low test proportions. Only the pre-trained model performs well in this scenario.

Further, it has already been mentioned in §5.5 that metathesis sound change is not being captured correctly by the network which requires further investigation. Overall, very few languages and language families are included in the data used. Thus, it is desirable to create such datasets for other languages with at least cognacy information to improve the unsupervised training firstly, which can be then employed in supervised training successfully with fewer training examples.

### Ethics Statement

The data and some modules of code used in this work are obtained from publicly available sources.

As stated in §4.4, the code for the model defined by Meloni et al. (2021) was not publicly available, hence we implemented it our own. Thereby the results produced by our implementation may slightly differ from those that would be produced by the original model. Otherwise, there are no foreseen ethical concerns nor conflicts of interest.

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

# Appendix

## A    Detailed results of CogTran-small on Reflex prediction task

| Family \Test prop. | 0.1 | 0.3 | 0.5 |
|---|---|---|---|
| Atlantic-Congo | 0.8192 | 0.7245 | 0.7027 |
| Hui | 0.7933 | 0.7418 | 0.7143 |
| Chapacuran | 0.6624 | 0.5847 | 0.5335 |
| Western Kho-Bwa | 0.8572 | 0.8065 | 0.7161 |
| Berta | 0.7681 | 0.6758 | 0.6197 |
| Palaung | 0.8815 | 0.7401 | 0.6863 |
| Burmish | 0.6531 | 0.5931 | 0.5261 |
| Indo-European | 0.5012 | 0.3915 | 0.3637 |
| Karen | 0.8869 | 0.7914 | 0.7504 |
| Bai | 0.8047 | 0.7029 | 0.6444 |

Family wise B-Cubed F scores for model CogTran-small against test proportions

## B    Detailed results of CogTran-small-pretrained on Proto-language reconstruction task

| Family \Test prop. | 0.1 | 0.5 | 0.8 |
|---|---|---|---|
| Romance | 0.7765 | 0.7570 | 0.7353 |
| Bai | 0.7465 | 0.7108 | 0.6748 |
| Burmish | 0.8426 | 0.7250 | 0.6460 |
| Karen | 0.8666 | 0.7845 | 0.7373 |
| Lalo | 0.7221 | 0.6769 | 0.6416 |
| Purus | 0.8941 | 0.8662 | 0.8440 |

Family wise B-Cubed F scores for model CogTran-small Pretrained against test proportions

| Test prop. | ED | NED | BCF |
|---|---|---|---|
| **0.1** | 0.065 | 0.015 | 0.018 |
| **0.5** | 0.028 | 0.006 | 0.008 |
| **0.8** | 0.027 | 0.006 | 0.007 |

Standard Deviations for model CogTran-small Pretrained