# OpenReview forum: "Cognate Transformer for Automated Phonological Reconstruction and Cognate Reflex Prediction"
_EMNLP/2023/Conference — EMNLP 2023 Main_

### Official Review · Reviewer_LYee · 2023-07-24

**Soundness:** 5

**Excitement:**

4: Strong: This paper deepens the understanding of some phenomenon or lowers the barriers to an existing research direction.

**Paper Topic And Main Contributions:**

This paper offers results using a method from computational biology (a kind of transformer model with 2d input) on supervised cognate reconstruction and protolanguage reconstruction.

Inputs are aligned, then presented to the transformer encoder, and then results are decoded using a classification head at each position. Results beat a 2022 shared task winner (cognate reflexes) and Meloni 2021 (protolanguage) which are good baselines.

There is some nice phonetic/phonological analysis.

**Questions For The Authors:**

Could you report scores per language family as an appendix?

Can you provide confidence intervals or statistical significance of numbers in tables?

**Reasons To Accept:**

The method seems effective, and importing novel ideas from biology is a good idea in this area.

This paper is meticulously documented and should be replicable.

The analysis section is useful in identifying further problems to solve in this area.

Beyond standard experiments, some additional manipulation of the percent of data held out for testing is also done, as well as zero-shot experiments which look at reconstruction using a supervised model but without in-family data.

---

Thanks for the additional data/analyses reported in the review responses.

**Reasons To Reject:**

I don't see any.

**Reproducibility:**

5: Could easily reproduce the results.

**Reviewer Confidence:**

4: Quite sure. I tried to check the important points carefully. It's unlikely, though conceivable, that I missed something that should affect my ratings.

---

> ### Author Rebuttal · Authors · 2023-08-28
>
> Thank you very much for your review. We are grateful for your feedback.
>
> Answering the queries, scores per language family can be verily made available in the Appendix. Standard deviations are provided here for the proto-language reconstruction task obtained from cross-validation for the best performing model CogTran-small-Pretrained.
>
> | Prop | ED | NED | B3_F1|
> |-------|------|--------|---------|
> | 0.1 | 0.06 | 0.02 | 0.02 |
> | 0.5 | 0.03 | 0.006 | 0.008 |
> | 0.8 | 0.03 | 0.006 | 0.007 |
>
> Since cognate-reflex-prediction was a SemEval task, cross-validation was not performed in order to directly compare with the results obtained in the competition.

---

### Official Review · Reviewer_8tWY · 2023-08-05

**Soundness:** 3

**Excitement:**

3: Ambivalent: It has merits (e.g., it reports state-of-the-art results, the idea is nice), but there are key weaknesses (e.g., it describes incremental work), and it can significantly benefit from another round of revision. However, I won't object to accepting it if my co-reviewers champion it.

**Missing References:**

Though the paper has citied most of the recent works in the task they lacks citations while defining the terms such as 'Phonological reconstruction, cognate reflex predictions.'

**Paper Topic And Main Contributions:**

The Paper tries to adapt the MSA transformer model to the problem of automated phonological reconstruction and cognate reflex prediction. The authors also claim that the model can also be pre-trained on the cognate reflex prediction tasks for the proto-language reconstruction. The paper reports experimental results on both tasks and compares them with the state-of-the-art models. It also shows an extensive error analysis report of the models.

**Questions For The Authors:**

Could please provide the t-test value of the experiments?

**Reasons To Accept:**

The idea of adapting the transformer model is good and innovative and would further. help in solving the tasks.
The paper provides a detailed error analysis report of the sound change error.
The experiment details are well documented and should be replicable.

**Reasons To Reject:**

To start off with the most problematic: The paper is highly similar in terms of description of the models, experimentation and writing style (List et al., 2022a). The model architecture is also not new and is heavily adapted from the previous work; However, although the authors have cited the paper, they still failed to provide reasons for the adaptation.

Some important facts that I would expect to be more clearly discussed were missing. In section 3.1, Multiple Sequence Alignment, the authors name the methodology and do not justify why they chose to do so. Citing the state-of-the-art does not justify why you are doing these steps in your experiment.
- The authors also mention that they input language information as a separate token attached to be beginning of the phoneme sequence, which is dissimilar to (Meloni et al., 2021). However, they do not provide any motivation for the steps or the drawback of (Meloni et al., 2021) steps as it is certain that they try to compare both models.
-Section 3.2 is poorly described. The authors should describe the reasons behind the decrease in performance of the trimming for testing.

**Reproducibility:**

5: Could easily reproduce the results.

**Reviewer Confidence:**

4: Quite sure. I tried to check the important points carefully. It's unlikely, though conceivable, that I missed something that should affect my ratings.

**Typos Grammar Style And Presentation Improvements:**

There are two spaces after the full stop on some of the pages.  Please recheck the two spaces and make them as one space uniformly.

---

> ### Author Rebuttal · Authors · 2023-08-28
>
> Thank you for giving an informative review with fine observations.
>
> Regarding the comment that the paper appears somewhat similar to (List 2022a), the pre-processing steps, namely, multiple sequence alignment, alignment trimming (Sec 3.1-2) and the dataset used are indeed same. For this reason and to save space, Sec 3.1 and 3.2 are not exclusively elaborated. Otherwise, sequence alignment and related procedures are quite common in this field and the reason behind can be intuitively understood by noting the similarity between the sound change phenomenon of historical linguistics and the phenomenon of genetic mutation. As pointed out and reiterated in the paper, the architecture is an adaptation from biology. Nevertheless, our work is the first to successfully apply a transformer-based architecture while obtaining SoTA results in this particular field.
>
> Regarding the lack of trimmings during testing, this is an empirical observation for which reason is not known.  Trimmings in the context of training applies mostly to the proto-language, i.e., gold output which often gets 'eroded' to give rise to the daughter languages. Since, proto-language form is unknown during testing phase, trimmings in this context would apply to daughter languages and as proto-languages are often not trimmed or 'eroded' forms of daughter languages, trimming is not only meaningless but also possibly brings down the performance.
>
> Regarding the deviations in the methodology from (Meloni et al. 2021), the transitions from it are motivated by the general trends in natural language processing. This is not only true with regard to the transition from BiGRU-Attention to attention only transformer architecture but also in discontinuing the usage of language embedding. The use of language embedding was found in the earliest multilingual large language model, XLM (Lample and Conneau 2019), but its usage discontinued in the later models. However, whether such step improves or decreases the performance is not verified in our case. Inputting language information as a token is convenient and less complex from an implementation perspective.
>
> Regarding the query to provide the t-test values, we report p-values obtained from paired t-test between the two best performing models on proto-language reconstruction, namely CogTran-small-pretrained and SVM + PosStr.
>
> | Prop | ED | NED | B3_F1 |
> |--------|------|--------|----------|
> | 0.1|0.60|0.60|0.90|
> |0.5|0.01|0.003|0.002|
> |0.8|2e-07|1e-08|3e-10|
>
> The improvements in scores become significant at high test proportions or low resource settings. Since cognate reflex prediction task was a SemEval task, cross-validation was not performed in order to directly compare the results from the competition.

---

### Official Review · Reviewer_ZH7n · 2023-08-08

**Soundness:** 3

**Excitement:**

2: Mediocre: This paper makes marginal contributions (vs non-contemporaneous work), so I would rather not see it in the conference.

**Paper Topic And Main Contributions:**

This paper proposes a transformer-based model for the tasks of supervised phonological reconstruction and cognate reflex prediction. The approach works by (1) using an out-of-the-box method for aligning phonemes within a cognate set, (2) trimming these alignments and collapsing phonemes accordingly, and (3) feeding the trimmed alignments into an MSA transformer architecture, which is a sequence alignment model from computational biology.

The approach is evaluated using a SIGTYP 2022 dataset as well as a protolanguage reconstruction dataset from List, et al. (2022a) spanning six different language families. The proposed model shows small gains over existing approaches, at least when pretrained on the cognate reflex prediction task.

**Questions For The Authors:**

- My impression is that most of the gains from this method come from pretraining on the cognate reflex prediction task rather than on the choice of model architecture. In particular, in Table 4, CogTran-small Pretrained seems to significantly outperform other methods, but CogTran-small and SVM + PosStr are roughly on par with each other. I’m not too familiar with the details of these other methods, but would it be similarly possible to apply the same type of pretraining approach to them? Is there a clear gain from the architecture used or is it mostly the pretraining that leads to performance improvements?
- Lines 521-524: Do you mean to claim that more than 40% of the reconstructions exactly match their original forms? This seems highly unlikely to me given the zero-shot setting, and I’m not sure how you get that number from the edit distance metrics.

**Reasons To Accept:**

- As far as I can tell, this model does provide small concrete improvements over previous modeling approaches, at least for the protolanguage reconstruction task (Table 4).
- Additionally, this paper brings in new methods from outside of NLP (in this case, computational biology). Doing so might expose readers to new techniques they aren’t yet familiar with

**Reasons To Reject:**

- The primary contribution of this paper is the application of an existing method (the MSA transformer) to a new domain (phonological reconstruction). While there is some evidence that this slightly improves upon existing methods, the novelty of the work is limited.
- Some of the improvements in edit distance (Tables 3-4) are reasonably small, and the paper would benefit either from significance testing on its results or from the inclusion of additional datasets. The paper briefly mentions cross-validation and significance testing for the proto-language reconstruction task but not for the cognate reflex prediction task (lines 448-450). The paper is also missing some details about how the significance testing works.

**In response to the rebuttal**: I have adjusted my soundness score (2->3) based on the author response to my concerns about significance testing.

**Reproducibility:**

3: Could reproduce the results with some difficulty. The settings of parameters are underspecified or subjectively determined; the training/evaluation data are not widely available.

**Reviewer Confidence:**

3: Pretty sure, but there's a chance I missed something. Although I have a good feel for this area in general, I did not carefully check the paper's details, e.g., the math, experimental design, or novelty.

**Typos Grammar Style And Presentation Improvements:**

- When discussing related work: it would be useful to draw a clearer distinction between supervised and unsupervised approaches to phonological reconstruction.
- Additional explanation of how the MSA transformer works would be appreciated, if space allows

---

> ### Author Rebuttal · Authors · 2023-08-28
>
> Thank you for giving an insightful review.
>
> As pointed out correctly, the novelty lies in transferring of an idea from one field to the other, which in this case, is from computational biology to computational historical linguistics. Our work is still the first, to the best of our knowledge, to successfully apply a transformer-based architecture while obtaining SoTA results in this particular field.
>
> Regarding the comment that improvements in score (Tables 3-4) are small, this is true for small test-train proportions. However, the increase is indeed significant for higher test proportions. We also stated that high-test-proportion scenarios are realistic or real-life ones (lines 425-430) since these are the cases where linguists get benefited the most by using automated procedures. It is in this low-resource setting that the improvements become significant. Supporting this statement, we report p-values obtained from paired t-tests between the two best performing models on proto-language reconstruction, namely CogTran-small-pretrained and SVM + PosStr.
>
>
> | Prop | ED | NED | B3_F1 |
> |--------|------|--------|----------|
> | 0.1|0.60|0.60|0.90|
> |0.5|0.01|0.003|0.002|
> |0.8|2e-07|1e-08|3e-10|
>
> The same test could not be carried out for the cognate reflex prediction task, as it was a SemEval task and cross-validation was not performed in order to directly compare the results from the competition. However, cross-validation can verily be incorporated for this task in future.
>
> Regarding the query of whether there is a clear gain from the architecture, the answer is yes but this is indeed unclear from Table 4 where SVM based approach performs similar or even slightly better than non-pre-trained CogTran-small. However, SVM based method was also part of cognate-reflex-prediction SemEval task where it lags significantly (see List 2022b), especially for high test proportions it is well behind the best models. Hence, there is clear gain from the architecture in terms of generalization. Regarding whether other methods, i.e., BiGRU + attention (Meloni et al. 2021) would benefit from pre-training, the answer is possibly so. However, as non-pretrained CogTran models are well ahead of non-pretrained BiGRU + attention (Table 4), it makes little sense to check if the pre-trained variant would bring any significant change in the overall ranking order amongst the models.
>
> Regarding the 40% accuracy mentioned in lines 521-524, the number is deduced from the normalized edit distance score, viz., 0.5758 ~ 60% (line 522). In other words, 40% of a word required no edits.

---

### Meta-Review · Area_Chair_VthY · 2023-09-18

**Recommendation:** 4

**Metareview:**

This paper applies a multiple sequence alignment (MSA) transformer designed to model protein sequences to the task of automatic phonological reconstruction. The authors find that their approach outperforms the best performing models from recent shared tasks on protoform reconstruction and cognate reflex prediction for a large number of language families. They extend their work with a qualitative error analysis, a discussion of zero-shot reconstruction, and a description of examples exploring how specific diachronic sound changes were learned by their models.

The reviewers appreciated the novelty of the idea of adapting a method from computational biology to a linguistic task. They also noted the improvements in performance over prior work reported for two distinct tasks with three different accuracy metrics. One reviewer had a very positive evaluation of this paper, finding no reasons to reject. Two reviewers, however, questioned the significance and novelty of the contribution.  Both reviewers seem to have misunderstood parts of the paper, despite the paper's overall clarity, and the author rebuttal addressed their concerns very thoroughly. Despite the middling scores from these two reviewers, the paper is clear and offers a substantial contribution that is entirely appropriate for this track.

---

### Decision · Program_Chairs · 2023-10-07

**Decision:**

Accept-Main

**Comment:**

This paper applies a multiple sequence alignment (MSA) transformer designed to model protein sequences to the task of automatic phonological reconstruction. The authors find that their approach outperforms the best performing models from recent shared tasks on protoform reconstruction and cognate reflex prediction for a large number of language families. They extend their work with a qualitative error analysis, a discussion of zero-shot reconstruction, and a description of examples exploring how specific diachronic sound changes were learned by their models.

The reviewers appreciated the novelty of the idea of adapting a method from computational biology to a linguistic task. They also noted the improvements in performance over prior work reported for two distinct tasks with three different accuracy metrics. One reviewer had a very positive evaluation of this paper, finding no reasons to reject. Two reviewers, however, questioned the significance and novelty of the contribution.  Both reviewers seem to have misunderstood parts of the paper, despite the paper's overall clarity, and the author rebuttal addressed their concerns very thoroughly. Despite the middling scores from these two reviewers, the paper is clear and offers a substantial contribution that is entirely appropriate for this track.